Resource

# Re-annotation of eight *Drosophila* genomes

Haiwang Yang[1] , Maria Jaime[1], Maxi Polihronakis[2], Kelvin Kanegawa[3], Therese Markow[4,2], Kenneth Kaneshiro[3], Brian Oliver[1]

**The sequenced genomes of the *Drosophila* phylogeny are a central resource for comparative work supporting the understanding of the *Drosophila melanogaster* non-mammalian model system. These have also facilitated evolutionary studies on the selected and random differences that distinguish the thousands of extant species of *Drosophila*. However, full utility has been hampered by uneven genome annotation. We have generated a large expression profile dataset for nine species of *Drosophila* and trained a transcriptome assembly approach on *D. melanogaster* that best matched the extensively curated annotation. We then applied this to the other species to add more than 10000 transcript models per species. We also developed new orthologs to facilitate cross-species comparisons. We validated the new annotation of the distantly related *Drosophila grimshawi* with an extensive collection of newly sequenced cDNAs. This re-annotation will facilitate understanding both the core commonalities and the species differences in this important group of model organisms, and suggests a strategy for annotating the many forthcoming genomes covering the tree of life.**

## Background

*Drosophila melanogaster* is a genetic and genomic workhorse that has led to the understanding of the chromosome theory of inheritance, the nature of mutations, pattern formation in development, innate immunity, circadian rhythms, and a host of other discoveries in the last century (Bilder & Irvine, 2017; Callaway & Ledford, 2017). There is a core set of 12 sequenced and assembled genomes in the *Drosophila* genus (Adams et al, 2000; Richards et al, 2005; Drosophila 12 Genomes Consortium et al, 2007; Hoskins et al, 2015). This is an important resource for studying diverse evolutionary biology problems, such as sex chromosome evolution (Charlesworth & Charlesworth, 2005), de novo gene formation (Lu et al, 2008), and duplication and divergence (Vieira et al, 2007). Using other *Drosophila* species for comparative genomics can also help identify the conserved genomic elements in *D. melanogaster*, in cases where frequent random occurrence obscures identification of DNA elements (Chen et al, 2014). For example, comparative genomics was a valuable tool for studying Doublesex DNA binding site function, as the short degenerate sequences bound by Doublesex appear by chance at a high rate (Clough et al, 2014). Comparative genomics is also essential for determining the probable function of transcribed elements. For example, short ORFs in "noncoding" RNAs are not commonly annotated because they occur often in a random sequence. But if a short ORF appears in a phylogeny, then those "noncoding" RNAs are likely to encode short biologically active peptides (Tautz, 2009).

The utility of the sequenced and annotated *Drosophila* genomes is clear, but there is room for improvement. The current annotations of non-*melanogaster* members of the genus are uneven and inferior to the heavily and actively curated *D. melanogaster* annotation (Gramates et al, 2017). For example, although there have been six versions of the *D. melanogaster* genome and upwards of 75 annotations (Hoskins et al, 2015), most the other species have a single assembly and one or two annotation versions (Richards et al, 2005; Drosophila 12 Genomes Consortium et al, 2007; Hu et al, 2013). Much of genome annotation depends on the identification of conserved long ORFs, but expression data presents a direct way to determine what portions of the genome are actively transcribed and should be annotated. There are transcript-driven annotation tools, such as Gnomon (Souvorov et al, 2010), but these are generic and not tuned to a particular lineage.

Dissected adult tissues are a good source for mRNAs to support genome annotation (Chintapalli et al, 2007; Chen et al, 2014). Although most tissues are present in both sexes, there are some sex-specific organs that show unique expression profiles (Arbeitman et al, 2002; Parisi et al, 2004; Graveley et al, 2011; Brown et al, 2014). For example, there are approximately 8,000 genes preferentially expressed in the testis and male reproductive tract and ~5,000 genes preferentially expressed in the ovary and female reproductive tract. In addition, because female transcripts maternally deposited in eggs are used during embryogenesis, many developmentally important transcripts are detected in adult female ovary samples. Overall, using dissected tissues from adults increases

[1]Section of Developmental Genomics, Laboratory of Cellular and Developmental Biology, National Institute of Diabetes and Digestive and Kidney Diseases, National Institutes of Health, Bethesda, MD, USA   [2]Drosophila Species Stock Center, Division of Biological Sciences, University of California San Diego, La Jolla, CA, USA   [3]Hawaiian Drosophila Research Stock Center, Pacific Biosciences Research Center, University of Hawai'i at Manoa, Honolulu, HI, USA   [4]National Laboratory of Genomics for Biodiversity (LANGEBIO), Irapuato, Guanajuato, Mexico

Correspondence: haiwangyang@gmail.com

coverage compared with whole samples, because of the fact that genes rarely expressed in a whole organism often show enriched expression in a given tissue (Chintapalli et al, 2007). More than 85% of annotated genes are expected to be covered in such experiments (Chintapalli et al, 2007; Daines et al, 2011). Although a full developmental profile, use of multiple genetic backgrounds, and environmental and/or genetic perturbations might marginally increase the coverage, these are less cost-effective than using adult tissues. In this work, we have used Poly-A$^+$ RNA-seq expression profiling of 584 samples from adults of *D. melanogaster* (*Dmel*), *Drosophila yakuba* (*Dyak*), *Drosophila ananassae* (*Dana*), *Drosophila pseudoobscura* (*Dpse*), *Drosophila persimilis* (*Dper*), *Drosophila willistoni* (*Dwil*), *Drosophila mojavensis* (*Dmoj*), *Drosophila virilis* (*Dvir*), and *Drosophila grimshawi* (*Dgri*) to support re-annotation of the corresponding genomes.

Performing de novo annotation based on gene expression is complicated by RNA coverage gaps that result in discontinuity within a single transcription unit, overlapping genes, and false splice junction calls due to gap generation that maximizes read alignment (Robertson et al, 2010; Sturgill et al, 2013). As a result, reference annotations and de novo transcript assemblies can differ radically (Garber et al, 2011; Haas et al, 2013). Some of these difficulties can be overcome, for example by using methods that capture strandedness (Grabherr et al, 2011). In addition, tuning the transcript parameters can improve the quality of the transcriptome (Vijay et al, 2013). These tools are often run using default parameters or based on some simple assumptions and tests. In this work, we decided to systematically test parameters and train support vector machines (SVM) on *D. melanogaster*, and then lift over these settings for automated annotation of the remaining species. This resulted in dramatic improvements in the mapping of RNA-seq reads to a greatly expanded set of genes and isoforms in these species. This general method might also be broadly applicable. If a few select species in the tree of life are targeted for heavily manual annotation, this can inform the automated annotation of the entire phylogeny.

# Results

## RNA-seq

*Dmel, Dyak*, *Dana, Dpse, Dper, Dwil, Dmoj, Dvir*, and *Dgri* represent a wide range of species separated by an estimated 40 million years of evolution (Leung et al, 2015), with fully saturated neutral substitutions at the widest separations (Fig 1A; after [Chen et al, 2014]). We targeted these genomes for re-annotation. We also included two *Dmel* strains: *w1118* and *Oregon-R* (*OreR*) to facilitate training the annotation. To evaluate the annotations of these nine members of the genus, we performed stranded Poly-A$^+$ RNA-seq experiments on quadruplicate biological samples derived from sexed whole flies and tissues (covering up to eight adult tissue types for each sex: whole organisms, gonads, reproductive tracts, terminalia, thoraxes, viscera, heads, and abdomens) for a total of 584 samples and ~5 billion RNA-seq reads (available at the Gene Expression Omnibus (GEO) [Edgar et al, 2002], under accession GSE99574 and a subset of accession GSE80124, see the Materials and Methods section).

## Annotation evaluation and optimized re-annotation

We first used RNA-seq data to evaluate the existing annotations. If an annotation was complete, the vast majority of our RNA-seq reads would map to annotated transcripts. On the other hand, if an annotation was poor, we would observe more RNA-seq reads mapping to unannotated regions and there might be extensive unannotated regions with mapped reads (Fig 1B). To determine how many reads aligning to the genomes mapped to annotated genes and transcripts, and how well those reads covered the existing models, we calculated the ratio of reads uniquely mapped to unannotated regions relative to the ones uniquely mapped to the whole genome by tissue type and species (Fig 1C). This metric was sensitive to read abundance from highly expressed unannotated genes. Therefore, in addition to the number of RNA-seq reads, we used a related metric describing the number of bases covered by at least one RNA-seq read outside of the annotations (Fig 1D).

We observed a wide range of annotation qualities using these simple metrics. For example, *Dmel* had only 1% of reads uniquely mapped to unannotated regions for all tissues except male reproductive tract, which had 2% mapping to unannotated regions, strongly suggesting that the *Dmel* genome annotation for highly expressed genes is nearly complete (Fig 1C). However, we still observed read alignment at up to 14% of unannotated *Dmel* regions (Fig 1D), suggesting that additional annotation is required, especially for fully capturing the transcriptomes of the testis, head, and thorax. In contrast, we observed that up to 46% of reads are uniquely mapped to unannotated regions in non-*melanogaster* species (Fig 1C). Similarly, we observed that up to 40% of unannotated regions had mapped read coverage (Fig 1D). *Dper* and *Dgri* had the poorest annotations. To determine if certain tissues might be especially valuable in completing the transcriptome, we also examined the reads mapping ratios by tissue. As we observed for *Dmel*, we found that RNAs from the ovary, reproductive tract, thorax, viscera, and abdomen had the best mapping, whereas RNAs from the testis and head of either sex had the poorest mapping to the annotations. In conclusion, our results suggested that all eight of the non-melanogaster annotations need major improvements to approach the quality of the *Dmel* annotation.

Given the clear superiority of the *Dmel* annotations, we decided to systematically develop de novo transcriptome annotations for the genus that would approximate the *Dmel* annotation quality. To determine the best method for generating these new annotations, we generated tens of thousands of de novo annotations of *Dmel*, where we systematically and iteratively honed-in on optimal transcript assembly algorithm parameters (nine StringTie parameters and >¼ million combinations), and used SVMs to develop filtering criteria such that we most closely matched the *Dmel* standard. We then used these settings and filters to generate new annotations for the remaining species (see the Materials and Methods section). We were gratified to find that in non-*melanogaster* species, the reads mapping to unannotated regions decreased fivefold after the annotation update (Fig 1E and F; number of reads in species and tissues before and after annotation update: Wilcoxon rank test, $P < 2.2 \times 10^{-16}$). As expected based on our use of *Dmel* as the "gold standard", there was no significant improvement in reads mapping exclusively to the annotation in *Dmel*. This

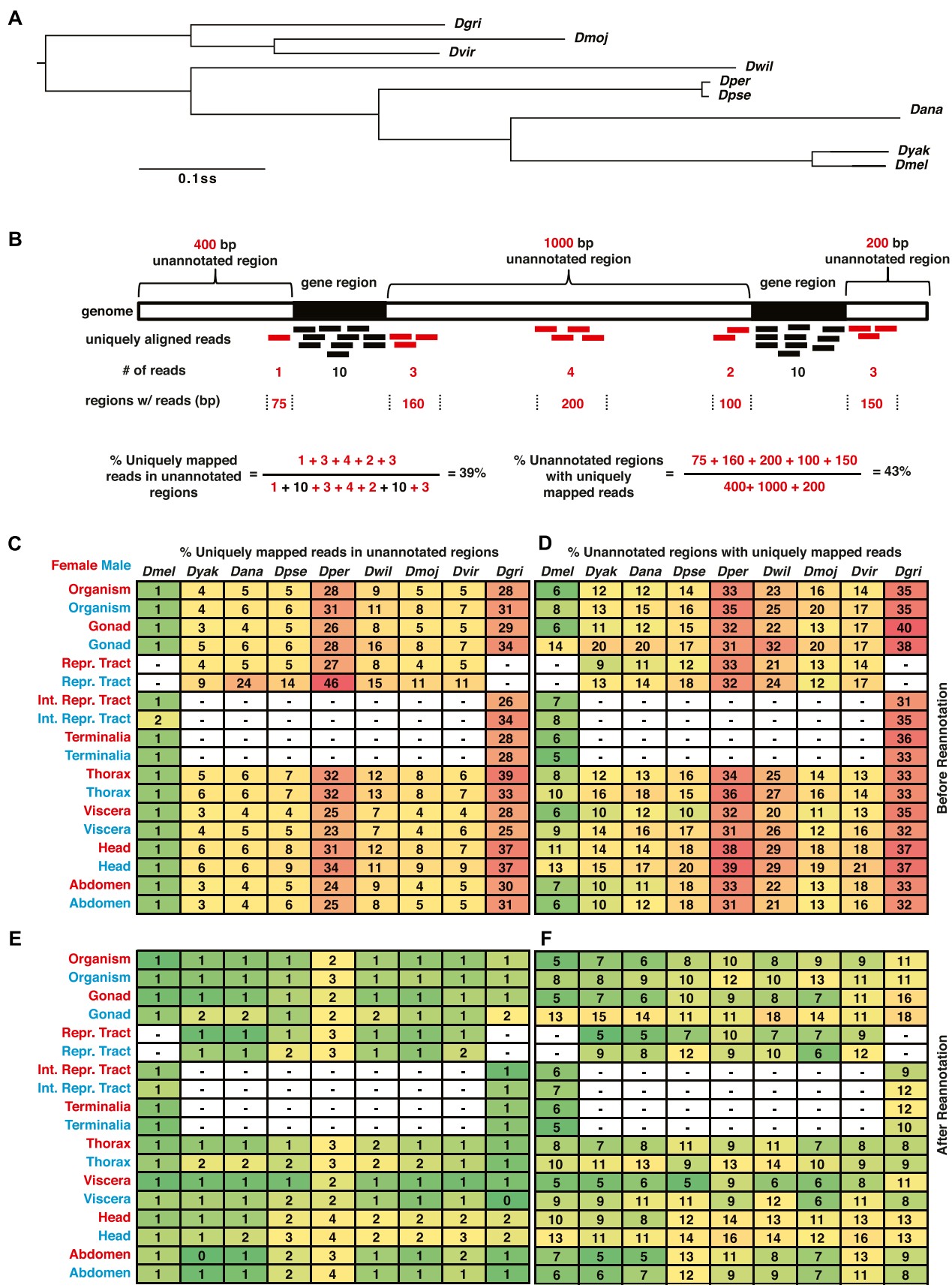

**A**

Dgri
Dmoj
Dvir
Dwil
Dper
Dpse
Dana
Dyak
Dmel

0.1ss

**B**

400 bp
unannotated region

gene region

1000 bp
unannotated region

gene region

200 bp
unannotated region

genome

uniquely aligned reads

\# of reads 1 10 3 4 2 10 3

regions w/ reads (bp) 75 160 200 100 150

$$\text{\% Uniquely mapped reads in unannotated regions} = \frac{1+3+4+2+3}{1+10+3+4+2+10+3} = 39\%$$

$$\text{\% Unannotated regions with uniquely mapped reads} = \frac{75+160+200+100+150}{400+1000+200} = 43\%$$

**C** % Uniquely mapped reads in unannotated regions

| Female Male | Dmel | Dyak | Dana | Dpse | Dper | Dwil | Dmoj | Dvir | Dgri |
|---|---|---|---|---|---|---|---|---|---|
| Organism | 1 | 4 | 5 | 5 | 28 | 9 | 5 | 5 | 28 |
| Organism | 1 | 4 | 6 | 6 | 31 | 11 | 8 | 7 | 31 |
| Gonad | 1 | 3 | 4 | 5 | 26 | 8 | 5 | 5 | 29 |
| Gonad | 1 | 5 | 6 | 6 | 28 | 16 | 8 | 7 | 34 |
| Repr. Tract | - | 4 | 5 | 5 | 27 | 8 | 4 | 5 | - |
| Repr. Tract | - | 9 | 24 | 14 | 46 | 15 | 11 | 11 | - |
| Int. Repr. Tract | 1 | - | - | - | - | - | - | - | 26 |
| Int. Repr. Tract | 2 | - | - | - | - | - | - | - | 34 |
| Terminalia | 1 | - | - | - | - | - | - | - | 28 |
| Terminalia | 1 | - | - | - | - | - | - | - | 28 |
| Thorax | 1 | 5 | 6 | 7 | 32 | 12 | 8 | 6 | 39 |
| Thorax | 1 | 6 | 6 | 7 | 32 | 13 | 8 | 7 | 33 |
| Viscera | 1 | 3 | 4 | 4 | 25 | 7 | 4 | 4 | 28 |
| Viscera | 1 | 4 | 5 | 5 | 23 | 7 | 4 | 6 | 25 |
| Head | 1 | 6 | 6 | 8 | 31 | 12 | 8 | 7 | 37 |
| Head | 1 | 6 | 6 | 9 | 34 | 11 | 9 | 9 | 37 |
| Abdomen | 1 | 3 | 4 | 5 | 24 | 9 | 4 | 5 | 30 |
| Abdomen | 1 | 3 | 4 | 6 | 25 | 8 | 5 | 5 | 31 |

**D** % Unannotated regions with uniquely mapped reads

| | Dmel | Dyak | Dana | Dpse | Dper | Dwil | Dmoj | Dvir | Dgri |
|---|---|---|---|---|---|---|---|---|---|
| Organism | 6 | 12 | 12 | 14 | 33 | 23 | 16 | 14 | 35 |
| Organism | 8 | 13 | 15 | 16 | 35 | 25 | 20 | 17 | 35 |
| Gonad | 6 | 11 | 12 | 15 | 32 | 22 | 13 | 17 | 40 |
| Gonad | 14 | 20 | 20 | 17 | 31 | 32 | 20 | 17 | 38 |
| Repr. Tract | - | 9 | 11 | 12 | 33 | 21 | 13 | 14 | - |
| Repr. Tract | - | 13 | 14 | 18 | 32 | 24 | 12 | 17 | - |
| Int. Repr. Tract | 7 | - | - | - | - | - | - | - | 31 |
| Int. Repr. Tract | 8 | - | - | - | - | - | - | - | 35 |
| Terminalia | 6 | - | - | - | - | - | - | - | 36 |
| Terminalia | 5 | - | - | - | - | - | - | - | 33 |
| Thorax | 8 | 12 | 13 | 16 | 34 | 25 | 14 | 13 | 33 |
| Thorax | 10 | 16 | 18 | 15 | 36 | 27 | 16 | 14 | 33 |
| Viscera | 6 | 10 | 12 | 10 | 32 | 20 | 11 | 13 | 35 |
| Viscera | 9 | 14 | 16 | 17 | 31 | 26 | 12 | 16 | 32 |
| Head | 11 | 14 | 14 | 18 | 38 | 29 | 18 | 18 | 37 |
| Head | 13 | 15 | 17 | 20 | 39 | 29 | 19 | 21 | 37 |
| Abdomen | 7 | 10 | 11 | 18 | 33 | 22 | 13 | 18 | 33 |
| Abdomen | 6 | 10 | 12 | 18 | 31 | 21 | 13 | 16 | 32 |

*Before Reannotation*

**E**

| | Dmel | Dyak | Dana | Dpse | Dper | Dwil | Dmoj | Dvir | Dgri |
|---|---|---|---|---|---|---|---|---|---|
| Organism | 1 | 1 | 1 | 1 | 2 | 1 | 1 | 1 | 1 |
| Organism | 1 | 1 | 1 | 1 | 3 | 1 | 1 | 1 | 1 |
| Gonad | 1 | 1 | 1 | 1 | 2 | 1 | 1 | 1 | 1 |
| Gonad | 1 | 2 | 2 | 1 | 2 | 2 | 1 | 1 | 2 |
| Repr. Tract | - | 1 | 1 | 1 | 3 | 1 | 1 | 1 | - |
| Repr. Tract | - | 1 | 1 | 2 | 3 | 1 | 1 | 2 | - |
| Int. Repr. Tract | 1 | - | - | - | - | - | - | - | 1 |
| Int. Repr. Tract | 1 | - | - | - | - | - | - | - | 1 |
| Terminalia | 1 | - | - | - | - | - | - | - | 1 |
| Terminalia | 1 | - | - | - | - | - | - | - | 1 |
| Thorax | 1 | 1 | 1 | 1 | 3 | 2 | 1 | 1 | 1 |
| Thorax | 1 | 2 | 2 | 2 | 3 | 2 | 2 | 1 | 1 |
| Viscera | 1 | 1 | 1 | 1 | 2 | 1 | 1 | 1 | 1 |
| Viscera | 1 | 1 | 1 | 2 | 2 | 1 | 1 | 1 | 0 |
| Head | 1 | 1 | 1 | 2 | 4 | 2 | 2 | 2 | 2 |
| Head | 1 | 1 | 2 | 3 | 4 | 2 | 2 | 3 | 2 |
| Abdomen | 1 | 0 | 1 | 2 | 3 | 1 | 1 | 2 | 1 |
| Abdomen | 1 | 1 | 1 | 2 | 4 | 1 | 1 | 1 | 1 |

**F**

| | Dmel | Dyak | Dana | Dpse | Dper | Dwil | Dmoj | Dvir | Dgri |
|---|---|---|---|---|---|---|---|---|---|
| Organism | 5 | 7 | 6 | 8 | 10 | 8 | 9 | 9 | 11 |
| Organism | 8 | 8 | 9 | 10 | 12 | 10 | 13 | 11 | 11 |
| Gonad | 5 | 7 | 6 | 10 | 9 | 8 | 7 | 11 | 16 |
| Gonad | 13 | 15 | 14 | 11 | 11 | 18 | 14 | 11 | 18 |
| Repr. Tract | - | 5 | 5 | 7 | 10 | 7 | 7 | 9 | - |
| Repr. Tract | - | 9 | 8 | 12 | 9 | 10 | 6 | 12 | - |
| Int. Repr. Tract | 6 | - | - | - | - | - | - | - | 9 |
| Int. Repr. Tract | 7 | - | - | - | - | - | - | - | 12 |
| Terminalia | 6 | - | - | - | - | - | - | - | 12 |
| Terminalia | 5 | - | - | - | - | - | - | - | 10 |
| Thorax | 8 | 7 | 8 | 11 | 9 | 11 | 7 | 8 | 8 |
| Thorax | 10 | 11 | 13 | 9 | 13 | 14 | 10 | 9 | 9 |
| Viscera | 5 | 5 | 6 | 5 | 9 | 6 | 6 | 8 | 11 |
| Viscera | 9 | 9 | 11 | 11 | 9 | 12 | 6 | 11 | 8 |
| Head | 10 | 9 | 8 | 12 | 14 | 13 | 11 | 13 | 13 |
| Head | 13 | 11 | 11 | 14 | 16 | 14 | 12 | 16 | 13 |
| Abdomen | 7 | 5 | 5 | 13 | 11 | 8 | 7 | 13 | 9 |
| Abdomen | 6 | 6 | 7 | 12 | 9 | 9 | 7 | 11 | 8 |

*After Reannotation*

harmonized annotation will provide an improved basis for comparative genomics studies. These annotations provide transcript-level features on par with those of *Dmel* (Table 1). We also provided annotation of all species as gtf/gff files in the GSE99574 and GSE80124 supplements so that researchers can make immediate use of this update.

### Summary of new gene models

Because the original annotation for the non-*melanogaster* members focused on conserved longest ORFs at each locus (Drosophila 12 Genomes Consortium et al, 2007), we anticipated that much of the improvement to the annotation would come from extending the annotation of UTRs and new isoforms because of alternative promoters, termination, and alternative splicing, as well as non-coding or minimally coding RNAs (ncRNAs). Indeed, we found that ~8,000 new gene models per species overlapped with and extended the older annotations (Fig 2A). For example, the *Dwil* gene *GK27243*, which is expressed in the testis, had the same splice junctions in the old and new annotation (*YOgnWI09161*), but had longer 5'- and 3'-ends in the updated annotation (Fig 2B). We also observed an increase of 10,000–20,000 isoforms in the new annotation compared with the old one (Fig 2C). For example, the *Dper doublesex* (*dsx*) locus (*GL23549*) had a single annotated isoform (Fig 2D). This is unlikely to be correct, as the *dsx* function is highly conserved (Yi & Zarkower, 1999) and the *dsx* locus encodes sex-specific transcription factors from sex-specifically spliced pre-mRNAs (Burtis & Baker, 1989). Our new annotation (*YOgnPE00925*) captured sex-specific isoforms of *Dper dsx* and includes a new upstream promoter. We also observed 700–1,300 instances per species where gene models were merged (Fig 2E). In at least some cases, this was strongly supported by expression data. For example in the case of the *YOgnWI03804* locus, the last two exons of *Dwil GK26840* are clearly joined by junction reads to the single exon of *GK20038* locus forming an updated gene model (Fig 2F). However, we did observe ~700 instances of merging in the well-annotated *Dmel* genome, which seems excessive. Overall, we generated 1,000–2,000 completely novel annotations per species (Fig 2G and H). These included ncRNAs (~24% of novel annotations), such as the *Dyak* noncoding homolog (*YOgnYA12879*) of *rna on X 1* (*roX1*). *Dmel roX1* is a component of the male-specific X-chromosome dosage compensation complex (Kuroda et al, 2016), and like the *Dmel* ortholog, the *Dyak roX1* locus is expressed in males, but not females. Loci-producing ncRNAs tend to diverge rapidly, but both the *Dmel* and *Dyak roX1* loci are flanked by the *yin* and *echinus* orthologs. The combination of sequence, expression pattern, and synteny strengthen the conclusion that these *roX1* genes descended from a common ancestral gene.

To identify other novel orthologs such as *roX1*, we analyzed the synteny, sex- and tissue-biased expression patterns, and gene

**Table 1. Number of features in the new annotations.**

|  | Genes | Transcripts | Exons |
|---|---|---|---|
| *Dmel*[a] | 17,730 | 35,105 | 83,251 |
| *Dyak* | 16,473 | 36,082 | 96,353 |
| *Dana* | 16,029 | 32,808 | 91,197 |
| *Dpse* | 16,441 | 39,527 | 103,772 |
| *Dper* | 17,726 | 35,392 | 97,224 |
| *Dwil* | 15,843 | 30,308 | 89,256 |
| *Dmoj* | 14,699 | 33,272 | 94,364 |
| *Dvir* | 15,074 | 33,357 | 93,631 |
| *Dgri* | 16,605 | 32,313 | 93,800 |

[a]Based on FlyBase annotation (release 2017_03).

structures of previously identified orthologs of *Dmel* genes and developed a SVM to generate a list of candidate orthologs, which were then compared at the sequence similarity level (see the Materials and Methods section). We conservatively called 500–1,000 new 1:1 orthologs per species (Fig 3A). They have on average two transcripts per gene and around three introns per transcript (Table 2). Their ratios of alignable regions with *Dmel* transcripts ranges from 0.69 to 0.81 and the ratio of identical bases ranges from 0.55 to 0.88, depending on the particular non-melanogaster species (Table 2). For example, we found orthologs of the noncoding *Dmel CR42860* gene in four of the eight other species (*YOgnYA06038*, *YOgnAN10714*, *YOgnWI07915*, and *YOgnVI13637* in *Dyak*, *Dana*, *Dwil*, and *Dvir*, respectively; Fig 3B). In each case, the *CR42860* ortholog is most strongly expressed in the thorax. Interestingly, in *Dwil*, *YOgnWI07915* also showed female-biased expression ($P_{adj} = 5.4 \times 10^{-10}$, DESeq2), highlighting the fact that we can observe changes in sex- and/or tissue-biased expression in the phylogeny. Extending the *Drosophila* orthology to include ncRNAs should allow for the exploration of conserved and divergent functions of this under-studied aspect of comparative genomics. In the future, a more formalized and generalized pipeline could be developed to extend this ortholog-finding methodology as a general tool.

### cDNA validation

Because we used the illumina RNA-seq data to build the new annotations, we needed an independent transcriptome dataset for validation. The *Dgri* annotation was greatly changed in our work and has been unexplored at the RNA-seq level. We therefore chose to validate a subset of the *Dgri* annotation. Our logic was that if we can validate the methodology using the most radically updated annotation, then the prospects for the rest of annotations are

**Figure 1. Evaluation of re-annotation for non-melanogaster *Drosophila* species.**
**(A)** Bayesian phylogenetic tree of nine *Drosophila* species. Nodes are supported by 100% posterior probabilities, and phylogenetic distance is shown as substitutions per site (ss). After (Chen et al, 2014). Standard three letter abbreviations are used (see text). **(B)** Cartoon of measurements of reads mapped to unannotated regions. Gene regions (fill), unannotated regions (open), RNA-seq reads (bars) mapped to gene regions (black) or unannotated regions (red) are shown. Numbers of reads (#), regions (in bps, flanked by dotted lines) with reads, and examples of calculating the ratios of unannotated reads are also shown. Percentage of reads **(D, F)** and regions with mapped reads **(D, F)** in unannotated regions before **(C, D)** and after **(E, F)** re-annotation. Green, yellow, and red denote low, median, and high percentage, which are also indicated in each cell. Expression profile tissues are shown (left) for female (red) and male (blue) samples. The reproductive (Repr.) tract includes the internal ducts and glands and the terminalia. The internal reproductive (Int. Repr.) tract does not include the terminalia. Missing tissues are shown (dash and open fill).

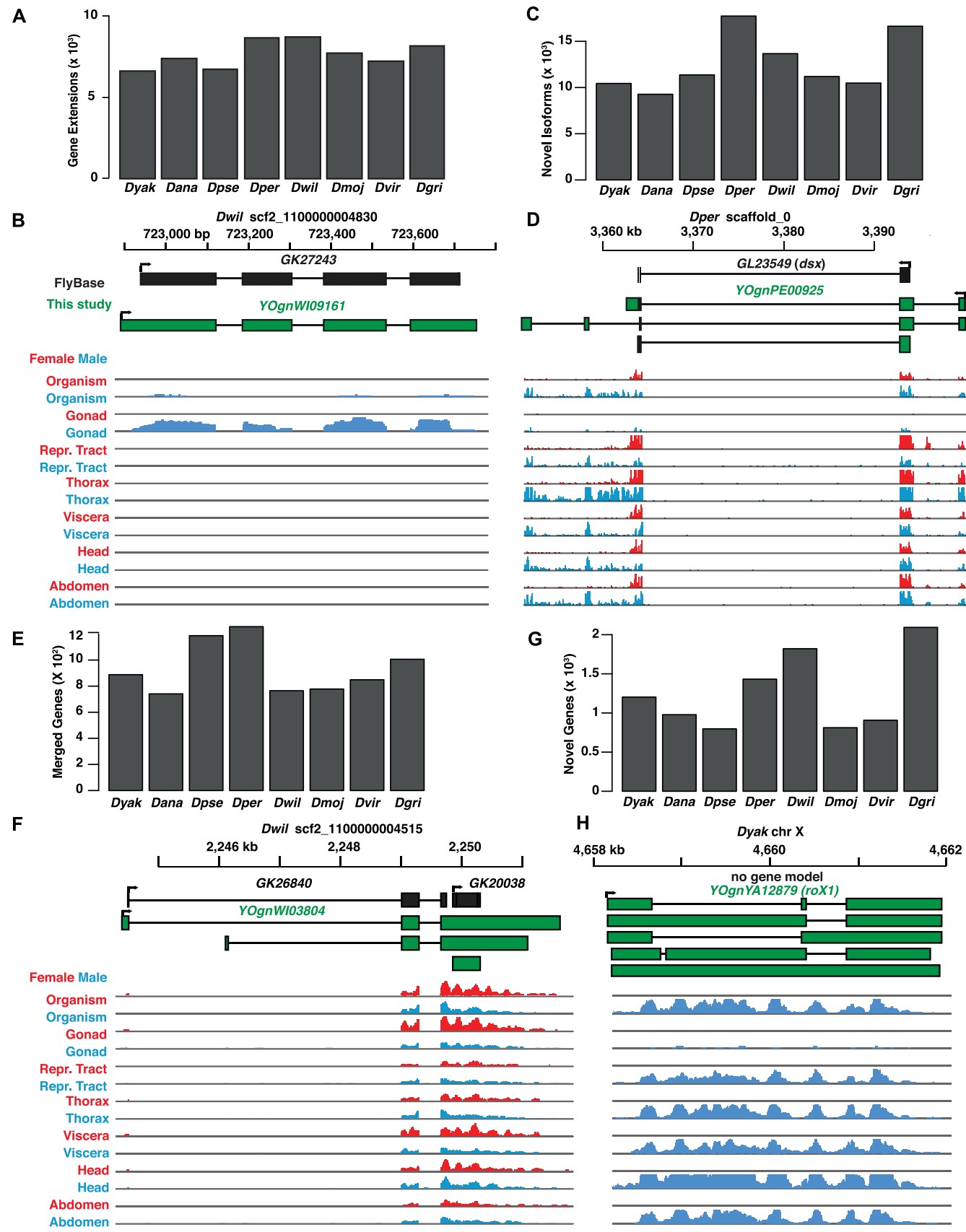

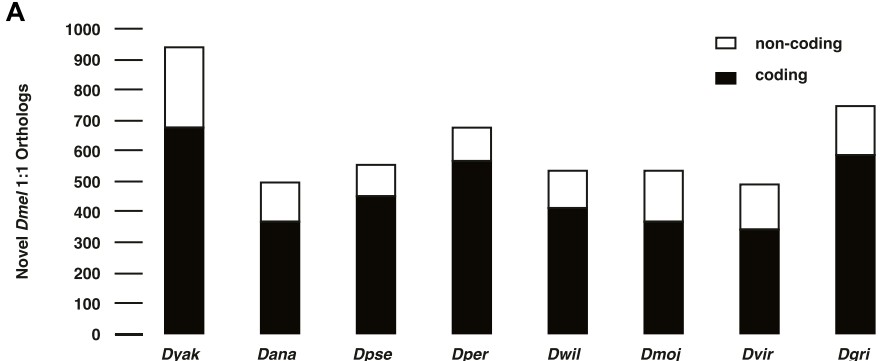

**A**

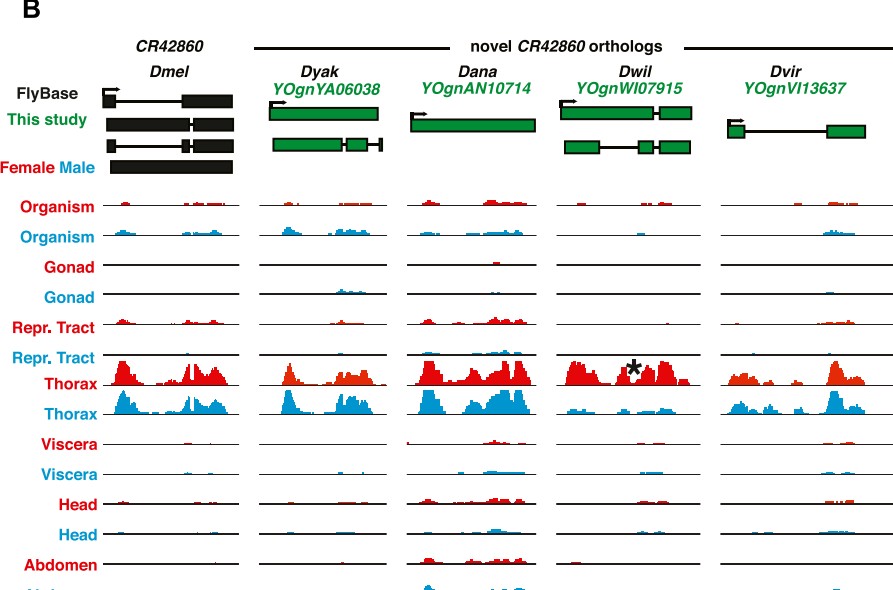

**B**

**Figure 3. Summary and novel 1:1 orthologs.**
**(A)** Number of novel *Dmel* 1:1 orthologs in each of the eight non-melanogaster species. The coding (filled) and noncoding (open) status of genes in *Dmel* are. **(B)** An example of novel 1:1 orthologs of *Dmel CR42860* identified in *Dyak*, *Dana*, *Dwil*, and *Dvir*. *CR42860* is nested in the intron of *sls* in all five species. Significant ($P_{adj}$ = 5.4 × 10$^{-10}$) female-biased expression of the *Dwil* ortholog is shown (asterisk). See Fig 2 for abbreviations and color coding.

outstanding. Longer reads have a distinct advantage for this validation, as they can capture the alternative splice forms without reliance on short junction reads. Therefore, we conducted duplicate PolyA+ PacBio Iso-seq for *Dgri* sexed adults. To better sample the transcriptome, we fractionated each RNA preparation into three size categories (see the Materials and Methods section). We generated 288,575 high-quality, full-length, non-chimeric, cDNAs and compared overlap with the annotation. This sampling experiment covered 27–29% of the FlyBase and new annotations (Fig 4A).

To systematically analyze the relationship between the cDNA sequences and the annotations, we measured the intersection/union (Jaccard index) for base coverage at the transcript isoform level (Fig 4B). The most dramatic difference in the Jaccard index was at zero, where there is no overlap between cDNAs and annotations.

We observed a dramatic decrease in the fully nonoverlapping cDNAs when we used the new annotation. The distribution of the Jaccard index scores shows a shift towards full overlapping, with 56% of cDNA/new annotations showing 75–100% similarity. For example, four cDNAs mapping to the *Dgri GH16482* locus (Jaccard index >0.94) validated the three isoforms of *YOgnGR06478*, with three distinct promoters, each of which has an extended 3'-UTR relative to the FlyBase annotation (Fig 4C). Similarly, we found support (Jaccard index >0.95) for the previously unannotated locus *YOgnGR03214* (Fig 4D).

In addition to providing transcriptome data from another independent platform, the PacBio cDNAs also provided us with an opportunity to evaluate the potential improvements that can be made to our updated annotation. Although an impressive 78% of

**Figure 2. Summary and re-annotation.**
**(A)** Number of new gene annotations with extended coverage and **(B)** one example. **(C)** Number of novel isoforms in the re-annotation and **(D)** one example. **(E)** Number of merged genes in the re-annotation and **(F)** one example. **(G)** Number of novel genes in the re-annotation, and **(H)** one example. **(A–H)** Species, scaffold IDs, and locus names are shown. FlyBase gene models (black) and new models (green) are shown. Orientation of transcripts are shown (arrow at 5'-end). Expression level tracks (arbitrary FPKM scale) for indicated tissues/sexes are shown.

**Table 2.** Features of novel 1:1 orthologs to *Dmel.*

|  | Transcripts per gene | Introns per transcript | Alignable/total | Identical/alignable |
|---|---|---|---|---|
| *Dmel*[a] | 2.0 | 4.3 | — | — |
| *Dyak* | 2.0 | 3.3 | 0.8 | 0.9 |
| *Dana* | 1.8 | 2.2 | 0.7 | 0.6 |
| *Dpse* | 2.4 | 3.8 | 0.7 | 0.6 |
| *Dper* | 2.3 | 2.6 | 0.7 | 0.6 |
| *Dwil* | 1.8 | 2.3 | 0.7 | 0.6 |
| *Dmoj* | 2.2 | 2.7 | 0.7 | 0.6 |
| *Dvir* | 1.8 | 2.2 | 0.7 | 0.5 |
| *Dgri* | 1.9 | 2.6 | 0.7 | 0.6 |

[a]Based on FlyBase annotation (release 2017_03).

288,575 cDNA are fully contained in our annotation, 24% had transcript ends beyond our annotation (5% with 5′ end extension, 4% with 3′ end extension, 14% with extension at both 5′ and 3′ ends, and <1% within intergenic regions) (Fig 4E). When looking at the 236,779 intron-containing cDNAs, we found that 87% had junctions compatible with our annotation (78% with all junctions present and 10% with called junctions annotated) and only 13% with novel junctions (1% with all junctions unannotated, 12% with junctions partially unannotated, <1% with antisense junctions, which are probably reverse transcription errors) (Fig 4F). Overall, these cDNAs suggest that the improvements in the annotations are substantial, but also highlight the ongoing need for annotation updates.

## Discussion

Previous work has used expression data in the *Drosophila* genus to validate the gene models in *Dmel* (Drosophila 12 Genomes Consortium et al, 2007; Chen et al, 2014), but there has been less systematic effort to use the knowledge from *Dmel* to inform and annotate the rest of the genus. To maximize the value of the sequenced *Drosophila* genomes, we have generated an extensive expression profile to assemble transcript models and update the annotations. In all cases, this resulted in an extensive set of new predicted transcripts. To leverage the decades of dedicated annotation that has been performed on *Dmel* (Adams et al, 2000; Celniker, 2000; Lewis et al, 2002; Misra et al, 2002; Celniker & Rubin, 2003; Drysdale, 2003; Ashburner & Bergman, 2005; Brown & Celniker, 2015), we generated thousands of de novo *Dmel* annotations, to determine optimal parameters and filters that resulted in the best match to the existing *Dmel* annotation. We then applied this optimized set of parameters to the rest of the genomes. This was largely successful, as we not only generated tens of thousands of new isoform models in each species, but we were also able to validate these models in *Dgri* with an independent set of full-length cDNAs. We suggest that when there is a high-quality annotation of a given species, this methodology could be used to tune the annotation pipelines for related species. The fact that this worked well for a species, *Dgri,* that is separated from *Dmel* by about 40 million years (Leung et al, 2015), suggests that targeting a few genomes in a lineage for full curation, can then result in a high-quality annotation for scores of related species.

The products of this article are the annotations. We hope that they are widely used by the evolutionary and comparative genomics communities. Given how often *Drosophila* are used in evolutionary studies on gene birth, death, duplication, and divergence, having improved annotations will facilitate a large number of studies. Similarly, workers interested in understanding *Dmel* genes should find these annotations useful for determining if a given feature is evolutionarily conserved. For example, this should be particularly true for identifying ncRNAs and RNAi-binding protein-binding sites in UTRs (Cech & Steitz, 2014), as those features were lacking for most non-*Dmel* genes. To make these annotations as useful as possible, we have posted the gene models and data in a number of formats (such as gtf, gff, bigWig, gene-level normalized read counts by DESeq2, and transcript-level Transcripts Per Million by Salmon). Some of these species will undergo reassembly in the future. The annotations for new assemblies will be relatively easy to update, without new experiments, as we saved all the unmapped reads (GSE99574 and GSE80124) at the sequence read archive (Leinonen et al, 2011).

There are efforts to sequence much of the tree of life, but manual annotation of those genomes would be prohibitively expensive. A more focused annotation effort on a key species in a phylogeny could be leveraged to annotate an entire phylogeny. We will need additional examples from other phylogenies to determine how sparsely high quality annotation will be needed, but within the species we examined here, our data suggests that organisms separated by tens of millions of years can have improved annotations using our method. The collective efforts from multiple groups in a genomics community could create reference annotations of large sets of genomes with a combination of focused effort on a single-type species and propagation of the annotation to the rest of the phylogeny.

## Materials and Methods

See the extensive reagent and resources table for strains, media, reagents and suppliers, software, database submissions, and other identifiers (Table S1).

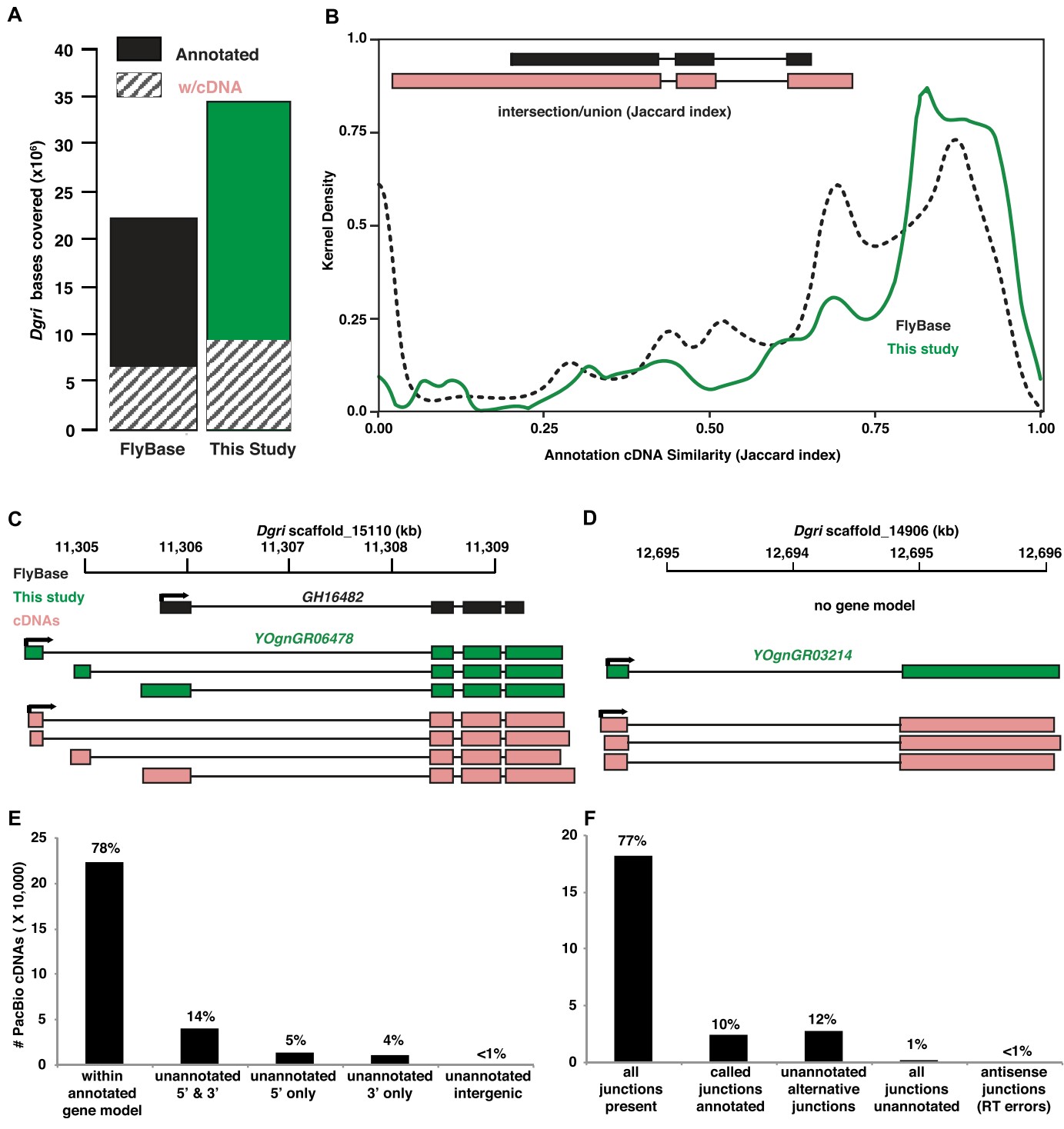

**Figure 4. PacBio cDNA validation of *D. grimshawi* annotation.**
**(A)** Bases in the genome covered by annotations (whole bars) and validated by PacBio Iso-seq cDNA sequencing (stripes) in *Dgri*. **(B)** Similarity between annotated gene models and cDNAs before (dotted black line) and after re-annotation (solid green line) as measured by the Jaccard index (intersection/union), as illustrated (inset). The Jaccard index distributions are shown in kernel density. **(C)** An example of cDNA (SRR6840922) aligning to a previous and new gene model, where the new gene models have extended coverage and a greater Jaccard index. **(D)** An example of cDNAs (SRR6840922) aligning with a novel gene model and a Jaccard score of "0" with the previous annotation. See Fig 2 for abbreviations and color coding. **(E)** Number and percentage of cDNAs within the annotated gene model, cDNAs with unannotated 5′ and 3′ ends, cDNAs with unannotated 5′ end only, cDNAs with unannotated 3′ end only, and cDNAs within the intergenic region. **(F)** Number and percentage of cDNAs with all junctions present in annotation (only intron-containing cDNA used in this analysis), cDNAs with called junctions annotated, cDNAs with unannotated alternative junctions, cDNAs with all junctions unannotated, and cDNAs with antisense junctions (probably because of reverse transcription error).

## Flies

Flies were grown at the National Institutes of Health (*Dmel*), the Drosophila Species Stock Center at University of California San Diego (*Dyak*, *Dana*, *Dpse*, *Dper*, *Dwil*, *Dmoj*, and *Dvir*), or the Hawaiian Drosophila Research Stock Center (*Dgri*). Growth conditions are given in GEO GSE99574 and GSE80124. For each sex and species, we dissected seven to eight adult tissues or body parts in PBS and transferred the samples immediately to RNAlater.

## RNA-seq

Illumina sequencing details are given in GEO (GSE99574 and GSE80124) and their GSM (i.e., sample accessions number) sub-pages. Briefly, we isolated RNA using the RNeasy 96 kit. We added External RNA Controls Consortium (ERCC) spike-ins (Jiang et al, 2011; Zook et al, 2012) for quality control purposes. We conducted single-end stranded 76-bp polyA$^+$ RNA-seq experiments for all samples using the TruSeq kit and protocol. We quantified nucleic acids with Quant-iT RiboGreen or PicoGreen kits. We multiplexed using both index adaptors and mixing RNA from distantly related species and sequenced on the HiSeq2000 Sequencing System. The use of species multiplexing has been used previously (Paris et al, 2015). De-multiplexed reads were produced by Illumina CASAVA (v1.8.2) as fastq files. We mapped the reads of mixed species libraries with HiSAT2 (v2.0.5; –dta and -max-intronlen = 300,000) and used SAMtools (v0.1.19) (Li et al, 2009) to sort the HiSAT2-generated bam files by read name. We used a python script demultiplexer (v1.0) to scan the bam file to collect the reads specific to one species or ERCC spike-ins (Jiang et al, 2011; Zook et al, 2012), and ambiguous reads, also provided in GEO. We converted from bam to fastq format using BEDTools (v2.25.0; bamtofastq) (Quinlan & Hall, 2010). We used the most current annotations of the species (FlyBase release 2017_03), except for *Dgri* because NCBI and FlyBase used our prepublication RNA-seq data to improve the annotation of *Dgri* using gnomon (Kapustin et al, 2008). We used the last version of *Dgri* annotation, before RNA-seq data inclusion (Flybase release 2016_05). The median Spearman's r of log-transformed gene expression among different biological replicates is above 0.95.

PacBio cDNA details are found at NCBI Sequence Read Archive (SRA) (SRP135764). Briefly, we constructed *Dgri* cDNA libraries following the isoform sequencing (Iso-Seq) protocol using the Clontech SMARTer cDNA Synthesis Kit and the SageELF size-selection system. We used 500 ng total RNA per reaction for the polyA+ enrichment and first strand synthesis, and conducted the first round of PCR amplification (95°C for 2 min; 14 cycles of 98°C for 20 s, 65°C for 15 s, 72°C for 4 min; 72°C for 5 min) to generate double-strand cDNA for size selection. We used three fraction ranges (SageELF index 10–12 or 1–2 kb, 8–9 or 2–3 kb, and 5–7 or 3–5 kb) of double-strand cDNA for the second round of PCR amplification (95°C for 2 min; N cycles of 98°C for 20 s, 65°C for 15 s, 72°C for X min; 72°C for 5 min). We repaired DNA ends and performed blunt-end ligation. We quantified SMRTbell libraries by Qubit Fluorometric Quantitation and qualified by Bioanalyzer beforesequencing on the PacBio RS II using DNA Sequencing Reagent kit 4.0 v2 with a run time of 240 min. We used sixth generation polymerase and fourth generation chemistry (P6-C4). Circular consensus (ccs2) reads

(–maxLength = 40,000 –minPasses = 1) were generated using PacBio pitchfork (v0.0.2) after conversion of bax.h5 files to bam using bax2bam. The final full-length nonchimeric Iso-seq reads were concatenated from three fractions and available in SRA (SRP135764).

## Annotation optimization

We developed a method to match the existing *Dmel* annotation with de novo RNA-seq data (Fig 5A). We mapped reads with HiSAT2 (v2.0.5; –dta and -max-intronlen = 300,000) (Kim et al, 2015). We then used StringTie (v1.3.3) (Pertea et al, 2015, 2016) to generate de novo annotation using the bam alignments from HiSAT2. We set minimum transcript length according to the shortest gene in *Dmel* (i.e., 30 bp), and we set the strandedness library to "--rf". We optimized "-c" (minimum reads per bp coverage to consider for transcript assembly), "-g" (minimum gap between read mappings triggering a new bundle), "-f" (minimum isoform fraction), "-j" (minimum junction coverage), "-a" (minimum anchor length for junctions), and "-M" (maximum fraction of bundle allowed to be covered by multi-hit reads). We used a union set of reads from replicated *w$^{1118}$* and *OreR* females and males to optimize StringTie parameters. To test which combination of parameters generated the de novo annotation closest to FlyBase, we used the Jaccard index (BEDTools v2.25.0) of unique exons to measure similarity. The use of Jaccard index is meant to eventually maximize the intersection between simulated annotation and FlyBase one and meanwhile minimize the union. In the first round of testing (Fig 5B), we tested all combinations of "c" (1, 3, 5, 7, 9), "g" (10, 30, 50, 70, 90), "f" (0.01, 0.03, 0.05, 0.07, 0.09), "j" (1, 2, 3, 5, 7, 9), "a" (5, 10, 15, 20, 25), and "M" (0.1, 0.3, 0.5, 0.7, 0.9). Among the 15,625 tests, the parameters with the highest Jaccard index were "c" = 3, "g" = 50, "f" = 0.01, "j" = 3, "a" = 10, and "M" = 0.9. In the second round of testing, we further picked all combinations of the points next to the previous optimal parameters with smaller intervals—"c" (1, 1.5, 2, 2.5, 3, 3.5, 4, 4.5, 5), "g" (30, 40, 50, 60, 70), "f" (0.005, 0.01, 0.015, 0.02, 0.025, 0.03), "j" (1, 2, 3, 4, 5), "a" (5, 6, 7, 8, 9, 10, 11, 12, 13, 14, 15), and "M" (0.7, 0.75, 0.8, 0.85, 0.9, 0.95). Among the 89,100 tests, the parameters with the highest Jaccard index were "c" = 1.5, "g" = 50, "f" = 0.015, "j" = 1, "a" = 14, and "M" = 0.95. In the third round of test, we further picked all combinations of the points next to the previous optimal parameters with smaller intervals—"c" (1, 1.1, 1.2, 1.3, 1.4, 1.5, 1.6, 1.7, 1.8, 1.9, 2), "g" (40, 41, 42, 43, 44, 45, 46, 47, 48, 49, 50, 51, 52, 53, 54, 55, 56, 57, 58, 59, 60), "f" (0.01, 0.011, 0.012, 0.013, 0.014, 0.015, 0.016, 0.017, 0.018, 0.019, 0.02), "j" (1, 2), "a" (13, 14, 15), and "M" (0.9, 0.91, 0.92, 0.93, 0.94, 0.95, 0.96, 0.97, 0.98, 0.99). Among the 152,460 tests, the parameters with the highest Jaccard index were "c" = 1.5, "g" = 51, "f" = 0.016, "j" = 2, "a" = 15, and "M" = 0.95. The above parameter optimization is compute intensive and a cluster computer with hundreds of central processing units is recommended. We applied the optimized parameters for all *Drosophila* species to generate sample-level de novo annotations. We then merged these sample-level annotations to species-level annotations for each species by StringTie (–merge). We optimized three parameters—"-F" (minimum input transcript FPKM), "-T" (minimum input transcript transcripts per million), and "-g" (gap between transcripts to merge together) using the same optimization (Fig 5C). The optimal parameter combination was F = 0, T = 10, and g = 0. We set minimum input transcript coverage ("-c") and minimum isoform fraction ("-f") as 1.5

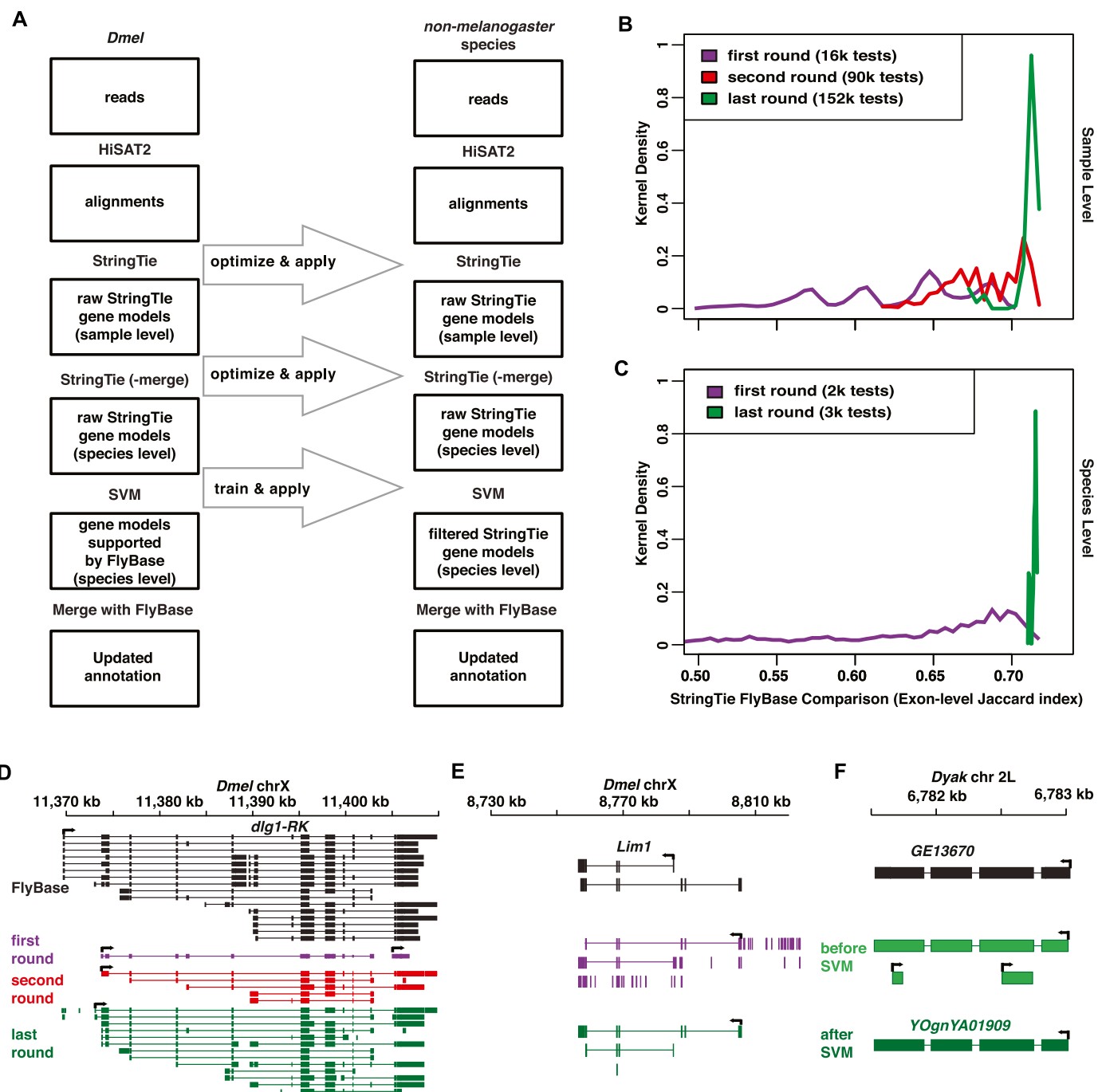

**Figure 5. Re-annotation and summaries.**
**(A)** Re-annotation. From top to bottom, HiSAT2 was used to map all RNA-seq reads back to the appropriate genome. StringTie was used to generate sample-level annotations and merging annotations to the species-level. SVMs were used to train the recognition of FlyBase gene models based gene features (e.g., exon length, intron length, and isoforms. see the Materials and Methods section). In each species, the new annotation was eventually merged with the corresponding FlyBase annotation by gffcompare to create the new annotation. The same optimized parameters in *Dmel* (left column) were applied in each of the non-melanogaster species (right column). **(B)** In *Dmel*, we progressively converged sample-level annotations on the FlyBase annotation in three rounds of parameter optimization in StringTie. **(C)** Again in *Dmel*, we permuted and tested species-level annotations to maximize similarity to the FlyBase annotation during parameter optimization in StringTie (-merge). **(B, C)** The X-axis is the exon-level Jaccard index between StringTie and FlyBase annotation. The Y-axis is the distribution of the Jaccard index scores in kernel density. First (purple), second (red), and last (green) rounds of optimization are shown. **(D)** An example of gene model improvements when generating sample-level annotations. **(E)** An example of gene model improvements when generating species-level annotations. Note that most gene models with short, single, and intron-less transcripts were removed in the last round of parameter optimization. **(F)** An example of applying SVM in a non-melanogaster species. FlyBase gene model was shown in black, and Gene models before (light green) and after SVM (dark green) are shown. See Fig 2 for abbreviations and additional color coding information.

**Table 3. Sensitivity and precision of StringTie annotation after SVM filtration at different levels.**

| | Base level | | Exon level | | Intron level | | Intron chain level | | Transcript level | | Gene level | |
|---|---|---|---|---|---|---|---|---|---|---|---|---|
| | S[a] | P[b] | S | P | S | P | S | P | S | P | S | P |
| *Dyak* | 78.4 | 85.8 | 61.2 | 70.6 | 73.9 | 89.5 | 44.5 | 52.9 | 43.4 | 51.2 | 57.0 | 70.0 |
| *Dana* | 78.1 | 85.5 | 59.7 | 70.6 | 71.9 | 89.7 | 45.3 | 53.9 | 44.2 | 52.7 | 55.8 | 70.5 |
| *Dpse* | 74.8 | 84.9 | 55.0 | 66.9 | 68.6 | 87.6 | 37.4 | 48.1 | 36.6 | 47.4 | 51.5 | 68.8 |
| *Dper* | 77.9 | 61.2 | 40.4 | 36.8 | 68.0 | 63.8 | 31.2 | 20.8 | 26.8 | 21.3 | 27.0 | 34.7 |
| *Dwil* | 80.7 | 68.8 | 51.1 | 49.3 | 73.5 | 74.8 | 44.3 | 33.2 | 40.7 | 33.1 | 42.6 | 50.6 |
| *Dmoj* | 79.1 | 83.1 | 59.4 | 65.3 | 73.1 | 86.2 | 44.2 | 46.2 | 43.1 | 45.9 | 54.7 | 67.9 |
| *Dvir* | 78.4 | 84.6 | 58.6 | 66.7 | 72.1 | 87.9 | 43.4 | 48.4 | 41.7 | 47.6 | 54.2 | 67.3 |
| *Dgri* | 79.2 | 59.1 | 44.5 | 37.6 | 74.3 | 66.2 | 37.3 | 23.4 | 31.0 | 22.8 | 31.2 | 38.8 |
| CV[c] | 0.02 | 0.15 | 0.14 | 0.25 | 0.03 | 0.13 | 0.12 | 0.32 | 0.17 | 0.32 | 0.25 | 0.26 |

[a]Sensitivity (%) = true positive/(true positive + false negative).
[b]Precision (%) = true positive/(true positive + false positive).
[c]Coefficient variation.
True positive: annotation identified in both StringTie and FlyBase.
False negative: annotation identified in FlyBase only.
False positive: annotation identified in StringTie only.
Values are obtained from gffCompare (with -r reference annotation).

and as 0.016, respectively, to be consistent with previous StringTie settings. The effect of optimization on output is illustrated in Fig 5D. We applied the optimized parameters for all *Drosophila* species to generate draft species-level de novo annotations.

To identify which StringTie-predicted gene models were already annotated in FlyBase, we used the Jaccard index to calculate the similarity of gene structure between StringTie gene models and FlyBase. We used a cutoff (Jaccard index >0.6), to obtain ~10,000 genes identified by both StringTie and FlyBase, and we deemed these genes as correctly predicted *Dmel* gene models of StringTie (Fig S1). However, we still observed an abundance of short single exon genes called by StringTie but not FlyBase (Fig 5E). To remove more of the StringTie unique calls, we used sequential and expressional features of genes to train a SVM (sklearn package v0.19.1 of Python v3.4.5) to recognize all the StringTie-predicted gene models with Jaccard index >0.6 with existing gene models in FlyBase. The features included isoform number, exon size, exon GC%, intron size, intron GC%, intron number (GT-AG intron type and other intron types, respectively), and median expression (DESeq2 normalized read counts) in 14 sexed tissues (except terminalia). For the SVM parameters, we tested different kernels (i.e., rbf, sigmoid), penalty parameter C (0.001, 0.01, 0.1, 1, 10, 100, 1,000), and kernel coefficient γ (0.001, 0.01, 0.1, 1, 10, 100, 1,000) values. The Receiver operating characteristic analyses indicated that the area under curve was the largest (0.97) at kernel = rbf and C = 10 and γ = 0.1. We also tested penalty parameter C (0.001, 0.01, 0.1, 1, 10, 100, 1,000) under the linear kernel (kernel coefficient γ is not available for this kernel), the maximum area under curve we obtained is 0.95, smaller than the optimal parameters. We applied the same SVM model with the optimized parameters to find qualified gene models in each species (Fig 5F). The sensitivity and precision of these annotations to FlyBase one were generated using gffCompare (with -r FlyBase annotation) and reported in Table 3. The drop of precision in *Dper*

and *Dgri* for all levels (Table 3) is consistent with the fact that these two species had the poorest annotations among all (Fig 1C and D).

To keep all StringTie gene candidates and FlyBase gene models in the updated annotation, we merged all qualified StringTie gene models with FlyBase annotation using gffCompare (v0.9.8) with option -r. In *Dmel*, we merged the StringTie gene candidates that were identified as correct prediction (i.e., Jaccard index >0.6 to FlyBase gene model) to the FlyBase annotation. If a StringTie transcript and a FlyBase transcript share the same structure for all introns on the same strand, we used the union of the gene structure of StringTie and FlyBase transcripts. After this step, the updated annotations were generated for each species. We used a universal ID format in the final updated annotations (e.g., *YOgnYA12345*). The format is Yang and Oliver (YO), gene (gn) or transcript (tr), species (*Dmel* (ME), *Dyak* (YA), *Dana* (AN), *Dpse* (PS), *Dper* (PE), *Dwil* (WI), *Dmoj* (MO), *Dvir* (VI), and *Dgri* (GR)), and a numerical identifier.

### Updated orthologs

We used gene synteny, tissue-level expression, splicing conservation, and sequential similarity relative to *Dmel* to search for new orthologs (Fig 6A) with random gene pairs as a null model. To determine flanking gene synteny, we first identified the flanking 10 genes on each side of the target gene determined by the best reciprocal blast hits using blastp (Altschul et al, 1990) of all the known protein sequences in FlyBase (Fig 6B). To compare the expressional similarity, we plotted the normalized read counts of 14 tissues (all except terminalia) in the species gene relative to the *Dmel* gene (Fig 6C). When comparing the gene structure similarity of any gene pair between *Dmel* and non-melanogaster, we calculated the ratio of (unique intron number + 1) between species gene and *Dmel* (Fig 6D). We used all known one-to-one orthologs between *Dmel* and non-melanogaster species to train an SVM

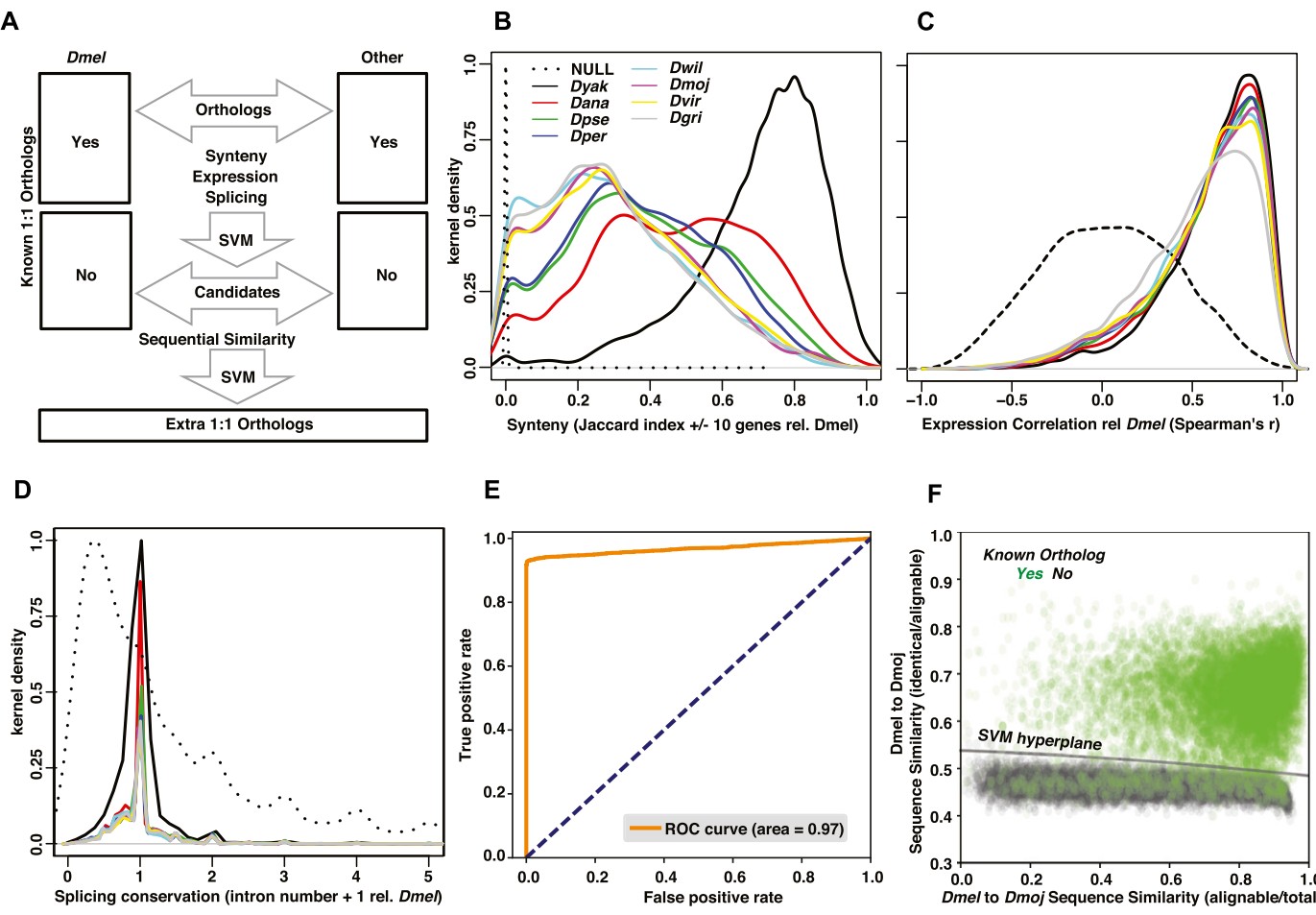

**Figure 6. Ortholog identification and summaries.**
**(A)** Pipeline to obtain extra 1:1 orthologs relative to *Dmel*. We used gene synteny, expression correlation among tissues, and exon structure similarity to train SVM models to recognize all known 1:1 orthologs. We used the same SVM model to generate more ortholog candidates among the genes that were not included in the current 1:1 ortholog dataset. Then we used sequence similarity (both alignable/total and identical/alignable) to finalize the extra 1:1 orthologs. **(B)** Distribution of orthologs in ± 10 gene window surrounding the query gene relative to *Dmel* for each non-melanogaster species (in kernel density). Solid lines are distributions of previously reported 1:1 orthologs, and dotted line (NULL group) is the expected distribution based on random gene pairs (generated by python random package) between *Dmel* and *Dyak* (the other non-melanogaster species all generate identical distributions) in genome (the same for the following panels). **(C)** Distribution of expression similarity among orthologs (Spearman's r) relative to *Dmel* for each non-melanogaster species (in kernel density). For each ortholog, normalized read counts of 14 sexed tissues between *Dmel* and each non-melanogaster species were used to calculated correlation. **(D)** Distribution of intron number relative to *Dmel* for each non-melanogaster species. Intron number plus one was used to avoid an infinite value. The different distribution in *Dyak* compared with that of the other non-melanogaster species in (B) to (D) is presumably because of its evolutionary closeness to *Dmel*. **(E, F)** An example of SVM training of sequence similarity between *Dmel* and *Dmoj*. Receiver operating characteristic curve (E) and visualization of SVM training (F). Known orthologs are shown as green dots, whereas random gene pairs are shown as black dots. The SVM hyperplane is shown as a solid line.

model to recognize possible ortholog candidates based on the above three features. The optimal parameter combination was rbf kernel, C = 1,000, and γ = 0.001. This procedure created a large number of potential orthologs. If there are multiple SVM predictions for one *Dmel* gene at this stage, we kept all these candidate orthologs. To be more conservative, we further filtered these ortholog candidates by using sequential similarity. We aligned the longest transcript of *Dmel* genes with that of non-melanogaster ortholog candidate by ClustalW (v2.1) with default parameters (Thompson et al, 1994). For each alignment, we calculated the ratio of alignable length to total length, and for the alignable region, we further calculated the ratio of identical length to alignable length. We used these two features to run SVM model (kernel = rbf, C = 1,000 and γ = 0.01) and selected the predicted orthologs (Fig 6E and F). If there are multiple SVM

predictions for one *Dmel* gene at this stage, we regarded them one-to-many orthologs and only used the one-to-one orthologs in the final results.

## Data Availability

All data resources and their locations were listed in the reagent table (Table S1). Briefly, the following is available at GEO under GSE99574 and GSE80124 (see specific GSM#s in Table S1): original RNA-seq reads (in fastq format); two demultiplexing versions, one based on HiSAT2 used here, and another generated by STAR; HTSeq raw read counts; gene-level DESeq2 normalized read counts; transcript-level expression from Salmon based on the updated

annotation; bigWig tracks for each tissue, sex, and species; updated annotations for the nine *Drosophila* species (including *Dmel*, which includes novel isoforms but not novel genes because of it being used as the training dataset) in both gff and gtf format. PacBio Iso-seq cDNAs are provided in the SRA (SRP135764). We also provided the updated 1:1 ortholog table between *Dmel* and non-melanogaster species. Junctions in bam format from HiSAT2 alignments of each sample (by SAMTools) are found at Zenodo (https://zenodo.org).

## Supplementary Information

## Acknowledgements

We are grateful to Ryan Dale, Justin Fear, Sharvani Mahadevaraju, Terence Murphy, Morgan Park, Harold Smith, Yijie Wang, and the Oliver lab for help and suggestions. The National Institute of Diabetes and Digestive and Kidney Diseases (NIDDK) Genomics core and the NIH Intramural Sequencing Center performed sequencing. This work utilized the computational resources of the NIH High-Performance Computing Biowulf cluster (http://hpc.nih.gov). This research was supported in part by the Intramural Research Program of the NIH, the NIDDK.

### Author Contributions

H Yang: conceptualization, data curation, formal analysis, visualization, methodology, and writing—original draft, review, and editing.
M Jaime: resources and data curation.
M Polihronakis: resources and data curation.
K Kanegawa: resources and data curation.
T Markow: resources and data curation.
K Kaneshiro: resources and data curation.
B Oliver: conceptualization, resources, data curation, formal analysis, supervision, funding acquisition, investigation, project administration, and writing—original draft, review, and editing.
BO and HY designed the experiment. MP, Kelvin Kanegawa, TM, and Kenneth Kaneshiro reared and maintained the flies. BO and HY dissected flies. HY and MJ performed RNA-seq library preparation. HY analysed the data. BO and HY wrote the manuscript. All authors read and approved the final manuscript.

### Conflict of Interest Statement

The authors declare that they have no conflict of interest.

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
