## [Reviewer comments · Life Science Alliance]

Reannotation of eight *Drosophila* genomes

Haiwang Yang, Maria Jaime, Maxi Polihronakis, Kelvin Kanegawa, Therese Markow, Kenneth Kaneshiro, and Brian Oliver

DOI: 10.26508/lsa.201800156

Review timeline:

Submission Date:	13 August 2018
Editorial Decision:	12 September 2018
Revision Received:	12 December 2018
Editorial Decision:	14 December 2018
Revision Received:	15 December 2018
Accepted:	16 December 2018

Report:

(Note: Letters and reports are not edited. The original formatting of letters and referee reports may not be reflected in this compilation.)

No Peer Review Process File is available with this article, as the authors have chosen not to make the review process public in this case.

1st Editorial Decision

12 September 2018

Thank you for submitting your manuscript entitled "Reannotation of eight *Drosophila* genomes" to Life Science Alliance. The manuscript was assessed by expert reviewers, whose comments are appended to this letter.

As you will see, all three reviewers think that your manuscript provides a valuable resource to others in the field and they all support publication of a slightly revised version in Life Science Alliance. The revision the reviewers expect is minor and straightforward to perform, so I am not listing the individual points here. Essentially, the individual requests all aim at strengthening the resource value of your work.

I would thus like to invite you to provide a revised version of your manuscript, following the constructive input provided by the reviewers. Should there be any revision point that needs further discussion, please get in touch with me.

Thank you for this interesting contribution to Life Science Alliance. We are looking forward to receiving your revised manuscript.

2nd Editorial Decision

14 December 2018

Thank you for submitting your revised manuscript entitled "Reannotation of eight *Drosophila* genomes". I appreciate the introduced changes and would be happy to publish your paper in Life Science Alliance pending final revisions necessary to meet our formatting guidelines.
